# Calcitriol Reduces the Inflammation, Endothelial Damage and Oxidative Stress in AKI Caused by Cisplatin

**DOI:** 10.3390/ijms232415877

**Published:** 2022-12-14

**Authors:** Beatriz M. Oliveira, Lucas Ferreira de Almeida, Amanda L. Deluque, Claudia S. Souza, Ana Lívia D. Maciel, Heloísa D. C. Francescato, Roberto S. Costa, Cleonice Giovanini, Francisco José A. de Paula, Terezila M. Coimbra

**Affiliations:** 1Department of Physiology, Ribeirao Preto Medical School, University of Sao Paulo, Ribeirao Preto 140490-900, Sao Paulo, Brazil; 2Department of Medical Clinic, Ribeirao Preto Medical School, University of Sao Paulo, Ribeirao Preto 140490-900, Sao Paulo, Brazil

**Keywords:** AKI, calcitriol, cisplatin, endothelium, CP-induced AKI, inflammatory process

## Abstract

Cisplatin treatment is one of the most commonly used treatments for patients with cancer. However, thirty percent of patients treated with cisplatin develop acute kidney injury (AKI). Several studies have demonstrated the effect of bioactive vitamin D or calcitriol on the inflammatory process and endothelial injury, essential events that contribute to changes in renal function and structure caused by cisplatin (CP). This study explored the effects of calcitriol administration on proximal tubular injury, oxidative stress, inflammation and vascular injury observed in CP-induced AKI. Male Wistar Hannover rats were pretreated with calcitriol (6 ng/day) or vehicle (0.9% NaCl). The treatment started two weeks before i.p. administration of CP or saline and was maintained for another five days after the injections. On the fifth day after the injections, urine, plasma and renal tissue samples were collected to evaluate renal function and structure. The animals of the CP group had increased plasma levels of creatinine and of fractional sodium excretion and decreased glomerular filtration rates. These changes were associated with intense tubular injury, endothelial damage, reductions in antioxidant enzymes and an inflammatory process observed in the renal outer medulla of the animals from this group. These changes were attenuated by treatment with calcitriol, which reduced the inflammation and increased the expression of vascular regeneration markers and antioxidant enzymes.

## 1. Introduction

Cisplatin (CP) is one of the most potent and effective anticancer drugs used in clinical practice [1]. Despite its effectiveness, its use is limited by its nephrotoxicity, which leads to kidney injury (AKI), and it has been shown that cisplatin accumulation in the kidney leads to selective damage to S3 of the proximal tubule located in the outer stripe of the outer medulla [2,3,4]. AKI is characterized by an abrupt drop in renal function, decline in glomerular filtration rate (GFR) and accumulation of metabolic waste [5]. Several mechanisms have been studied to evaluate the determinants for the nephrotoxic effect of CP. The kidney’s toxicity from cisplatin has been associated with the basolateral uptake along the proximal tubule via the organic cation transporter 2 (OCT2), leading to an intracellular cisplatin concentration up to five times higher than that of plasma levels [6].

This accumulation can lead to an increased production of reactive oxygen species (ROS) [7,8], triggering oxidative stress [9], vascular injury [10] and activation of inflammatory pathways [11] and apoptotic pathways [11,12].

To maintain cellular homeostasis, a balance between ROS production and antioxidant defense activity is necessary [13,14]. CP interferes with this balance by increasing the production of ROS [15] and reducing the production of antioxidant enzymes, such as superoxide dismutase (SOD) and catalase (CAT) [16]. The generation of oxidative stress results from this imbalance [17]. Oxidative stress caused by CP has been associated with endothelial damage that results in increased production of ROS which interferes with the activity of vasoactive substances [18].

The role of inflammation in the pathogenesis of CP-induced AKI has already been demonstrated by several studies [11,19,20,21,22]. CP activates the expression of critical proinflammatory cytokines, such as IL-1β, which is crucial for responses to infection and injury [23]. IL-1β is expressed in a wide range of tissues and various cells, especially in macrophages during the inflammatory process. It is also expressed in the glomeruli, in the outer cortical areas of the kidney and in many specific cell types, including endothelial and epithelial cells, fibroblasts and smooth muscle cells [24,25].

The active form of vitamin D (calcitriol or 1,25-dihydroxy vitamin D_3_), a fat-soluble nutrient, is known for regulating the metabolism of calcium and phosphorus, essential factors in bone remodeling [26]. Studies have also shown that calcitriol acts through the vitamin D receptor (VDR) of most cells. It can regulate the transcription of over 200 genes and directly or indirectly influence cell proliferation, differentiation and the immune system [27]. Calcitriol is vital in maintaining the structure and cell integrity and preserving renal function [28]. In addition, our research group showed that calcitriol played an essential role in the oxidative damage caused by AKI in a rhabdomyolysis model by decreasing oxidative damage markers [29].

In the present study, we explored the effects of calcitriol administration on the pathophysiology of cisplatin-induced AKI that involved four main mechanisms: (1) proximal tubular injury, (2) oxidative stress, (3) inflammation and (4) renal vascular damage. We hypothesized that treatment with calcitriol could modulate or even attenuate activation of these mechanisms, protecting the kidney from changes in function and structure induced by CP.

## 2. Results

### 2.1. Studies of Renal Function

Rats injected with cisplatin showed increased levels of plasma creatinine (Pcreat), fractional excretion of sodium (FENa+) and decreased glomerular filtration rate (GFR) when compared to control groups (SAL and SAL + calcitriol). The CP group treated with calcitriol showed an improvement in renal function (Table 1).

### 2.2. Cisplatin Provoked Renal Injury and Inflammation That Was Ameliorated by Calcitriol Treatment

The typical histological features of CP-induced AKI (characterized by loss of the brush border, cell necrosis, tubular dilation, sloughing and obstruction) and a higher tubular damage score were observed following the CP treatment. Nevertheless, these abnormalities were markedly reduced by treatment with calcitriol (CP + calcitriol group) (Figure 1A–E).

The epithelial cells express vimentin before differentiation or in the transdifferentiation processes that occur through a process known as the epithelial–mesenchymal transition (EMT) [30]. During this process, these cells can proliferate, migrate and produce an extracellular matrix. Therefore, this protein can be used as a marker of cellular damage [27,28]. The immunohistochemical analysis showed an increase in the expression of vimentin in the renal outer medulla in the animals in the CP group compared to the control groups (SAL and SAL + calcitriol) (Figure 2A–D). Treatment with calcitriol decreased vimentin expression in the renal outer medulla of these animals (Figure 2I–K). The tubular injury caused by CP was also evaluated through the cell proliferation. The number of PCNA-positive cells was analyzed in the renal outer medulla (Figure 2E–H). PCNA is an antigen in the nucleus of cells present in the proliferation phase that are indicative of recent tubular injury. The number of PCNA-positive cells was increased in the renal outer medulla of the animals injected with CP compared to the control groups (SAL and SAL + calcitriol), showing an intense tubular lesion in these animals. However, such alterations were attenuated by treatment with calcitriol (Figure 2H–L).

An inflammatory process evidenced by a large infiltration of macrophages (ED1-positive cells) in the tubulointerstitial of the outer medulla from the kidneys in the cisplatin-treated rats was also observed (Figure 3A–D). Calcitriol treatment prevented macrophage infiltration in this tubulointerstitial area (Figure 3E). Analysis of IL-1β expression in renal tissue showed higher levels of this cytokine in the CP group than in the controls (Figure 3F), which was attenuated by calcitriol treatment. Interleukin-10 (IL-10), a cytokine with anti-inflammatory and immunomodulatory functions, showed a downregulation in the CP group, while in the calcitriol group, upregulation was observed (Figure 3G).

### 2.3. Evaluation of 25-OH Vitamin D, VDR and the Cubilin Receptor

There was no significant difference between groups either concerning the mean serum concentration of 25-OH vitamin D or the classification of levels as deficient (levels < 20 ng/mL) or sufficient (levels > 30 ng/mL) (Figure 4F). However, the Western blot analysis demonstrated reduced VDR expression in renal tissue from the CP group compared with the control groups (SAL and SAL + calcitriol) (Figure 4G,H). The reduction in the expression of this protein induced by CP was attenuated by treatment with calcitriol. We also observed in immunohistochemical analysis that the number of tubules with a brush border marked with cubilin receptors was smaller in the CP group compared with the control groups (SAL and SAL + calcitriol) (Figure 4A–D). Calcitriol treatment attenuated these alterations (Figure 4E).

### 2.4. The Endothelial Damage Induced by CP Was also Improved by Calcitriol

The immunohistochemistry studies with JG12, a marker of endothelial cells, showed that CP animals had fewer capillaries in the renal cortex and outer medullae (Figure 5A–D). The effect of CP on the renal endothelium, evaluated by Western blot using a specific marker for endothelial cells (CD34), showed that the CD34 expression was reduced in the CP groups compared to the control groups (SAL and SAL + calcitriol) (Figure 5F). The calcitriol treatment improved these alterations (Figure 5E,G). Western blot analysis also demonstrated a reduction in tissue NOS3 (Figure 5H,I) and p-NOS (Figure 5J,K) expression in the CP group compared to the CP + calcitriol group.

Vascular endothelial growth factor (VEGF) is a significant proangiogenic factor in angiogenesis. We observed that the expression of VEGF was reduced in the animals of the CP group compared with that in the control rats (SAL and SAL + calcitriol) (Figure 6A,B). This alteration was reversed by treatment with calcitriol, evidenced by the increase in the expression of VEGF in the CP + calcitriol group. VEGF exerts its actions through its VEGFR2 receptor on target cells. We observed that those in the CP + calcitriol group also showed an increase in the expression of VEGFR2 in the kidney tissue (Figure 6C,D).

CXCR4 is a receptor present on endothelial cells and pericytes of hypoxic tissues. We observed that the animals in the CP group showed an increase in the renal expression of this receptor compared to controls (SAL and SAL + calcitriol) (Figure 6E,F). This increase in expression was attenuated in the CP + calcitriol group. Reducing antioxidant enzymes is one of the mechanisms of generating oxidative stress triggered by CP, and the increase in oxidative stress leads to endothelial damage. To assess the participation of this mechanism, we analyzed the expression of the antioxidant enzyme EC-SOD in kidney tissue (Figure 6F–H). We observed that CP reduced the expression of EC-SOD, and the antioxidant action of calcitriol was confirmed by the increased expression of EC-SOD in the SAL + calcitriol group. This action was also observed in the animals of the CP + calcitriol group, evidenced by the maintenance of EC-SOD levels in this group.

## 3. Discussion

The data presented in this manuscript provide evidence that calcitriol can attenuate inflammation, endothelial injury, oxidative stresses and epithelial cell injury in cisplatin-induced AKI. In addition, our study demonstrates that calcitriol also may display an anti-inflammatory action through modulation of IL-1β and IL-10, leading to a downregulation of IL-1β and upregulation of IL-10. Consistent with our data, previous studies have shown that cisplatin (CP) nephrotoxicity was associated with increased expression of IL-1β [31,32,33], and its inhibition alone does not protect against CP-induced AKI [31]. A recent study revealed that renal tubular epithelial cell-derived IL-1β polarizes renal macrophages toward a proinflammatory phenotype that stimulates salt sensitivity through the increase of renal IL-6 [34]. IL-10 is a cytokine known for its anti-inflammatory actions [35] and is produced by many immune cells [36,37]. Its anti-inflammatory actions are attributed to its ability to inhibit the infiltration of monocytes and neutrophils and the production of inflammatory cytokines [38,39,40]. A reduction in IL-10 levels has already been observed in CP nephrotoxicity [31]. Its increased expression has been suggested to be protective against CP-induced kidney injury [41,42]. Amirshahrokhi et al. (2015) observed that the reduction in renal toxicity caused by CP might be related to inhibiting proinflammatory cytokines [32]. Our results showed that this exact mechanism could be involved in calcitriol’s downregulation of IL-1B and reduced inflammation. In addition, vitamin D induces an increase in IL-10 secretion by regulatory T cells, an effect observed in a study with a patient with systemic sclerosis [43].

The increased vimentin expression was also observed in tubular cell injuries in the renal outer medulla from CP-injected rats. Tubular cells only express vimentin when proliferating, demonstrating recent lesions of these cells. The increased number of PCNA-positive cells confirmed this result. Calcitriol treatment decreased tubular cell injury and the expression of vimentin and PCNA in the animals injected with CP. Tan et al. (2006) observed that treatment with paricalcitol (a synthetic vitamin D analog) significantly reduced the expression of PCNA and attenuated renal interstitial fibrosis in a model of obstructive nephropathy [44]. The authors also observed that vitamin D treatment restored the expression of the VDR receptor, blocked epithelial–mesenchymal transition and inhibited cell proliferation, demonstrating that vitamin D plays a protective role in cellular integrity against this cell injury process.

Previous results from clinical and animal studies have suggested that VDR activation has beneficial effects on various renal diseases [45,46]. To address this question, we examined the VDR in kidney tissue and found a lower expression in the animals from the CP group, while the CP group that received calcitriol showed an increase in its expression.

In the present study, we observed that the lesions in the renal outer medulla were associated with decreased cubilin receptor expression in the apical region of the tubule cells in the CP group. The reduction in the number of tubules expressing cubilin in the cell brush border could lead to disturbances in vitamin D activation [47,48]. Our results showed that calcitriol-treated rats present preserved cubilin receptors, demonstrating the renoprotective role of calcitriol. Additionally, studies have shown that vitamin D deficiency is a risk factor for contrast-induced AKI due to an imbalance in intrarenal vasoactive substances and oxidative stress [49]. Our results showed that decreased expression of EC-SOD (an antioxidant enzyme) induced by CP was attenuated by calcitriol treatment. A similar effect was observed by Li et al. (2017), where pretreatment with cholecalciferol, an inactive form of vitamin D3, partially protected against ischemia-reperfusion-induced AKI through regulation of oxidant enzymes and suppression of oxidative stress [50].

CP-induced nephrotoxicity is due in part to vascular damage and the vasoconstriction associated with endothelial dysfunction and abnormal vascular self-regulation [51]. Vascular injury results in decreased renal blood flow and GFR, causing hypoxic tubular damage [52]. We have previously shown the participation of the endothelium in the development of AKI induced by CP [53].

We used JG12 to assess changes in capillary density in the outer medulla of the kidney. JG12 is a specific marker for the blood vessel endothelium and discs, and is instinctively expressed by the endothelial cells of tubulointerstitial vessels in the kidney [54,55]. Using a unilateral ureteral obstruction model, Sun et al. (2012) observed an alteration in peritubular capillary density detected using JG12 immunostaining [56]. In the present study, we observed that JG12-positive peritubular capillaries were markedly diminished from the outer medulla regions with significant interstitial expansion and tubular atrophy.

Following the loss of the peritubular capillaries, the CP group also presented with decreased production of NOS3 and p-NOS. In the vascular endothelium, NOS3, which is also known as nitric oxide synthase (eNOS), is an enzyme that produces NO. The decreased p-NOS expression in the renal tissues in our study may be due to decreased NOS3 expression, which can lead to increased vasoconstriction and contribute to alterations in blood pressure [57].

Renal vasculature quiescence is tightly regulated by the balance between pro- and antiangiogenic factors in healthy kidneys. However, this quiescence can be disrupted during AKI, resulting in an antiangiogenic environment with the loss of peritubular capillaries [58]. The increase in the expression of VEGF and VEGFR in the calcitriol group reinforces the role of calcitriol as an endothelium promoter since several studies demonstrate that VEGF promotes the growth of endothelium and protects the endothelial cells from apoptosis.

We next evaluated the effects of CP on CXCR4 expression in the kidney tissue. The increased expression of CXCR4 observed in the CP group could also contribute to epithelial and endothelial cell damage. In recent work, Chang et al. (2021) observed that suppression of the SDF-1/CXCR4 pathway resulted in increased tubular cell regeneration and reduced cell death and attenuation of microvascular rarefaction in IR-AKI mice kidneys [59]. This finding is consistent with our data, which show that calcitriol suppressed CXCR4 expression in the CP + calcitriol group, which was followed by the amelioration of endothelium and epithelial cell dysfunction.

In this study, we demonstrate that calcitriol attenuates the morphological and functional changes that occur in CP-induced AKI. However, some limitations must be recognized. First, studies through in vitro experiments using primary kidney endothelial cells and angiogenesis assays should be performed to better evaluate these events on the renal microvasculature. Second, we measured plasma creatinine levels with routine clinical laboratory methods but not with high-performance liquid chromatography, a more reliable method for evaluating plasma creatinine levels. Third, studies are needed to evaluate how calcitriol can reduce inflammatory cell infiltration and epithelial cell proliferation and accelerate the resolution and repair of epithelial cell injury. Despite these limitations involving experimental models, the present study adds to the literature concerning the participation of calcitriol administration on proximal tubular injury, oxidative stress, inflammation and vascular injury observed in CP-induced AKI.

In conclusion, our study suggests that calcitriol attenuates tubular injury, endothelial damage, reductions in antioxidant enzymes and the inflammatory process observed in the renal outer medulla observed in CP-induced AKI.

## 4. Materials and Methods

### 4.1. Animal Model and Experimental Design

The protocols were performed by the Animal Experimentation Committee of the University of São Paulo at the Ribeirao Preto Medical School (COBEA/CETEA/FMRP-USP, protocol no. 115/2018). This study used male Hannover rats (200–300 g). The rats were housed four per cage according to the groups, with a room temperature of 22 ± 2 °C, a 12 h light/dark cycle with a chow diet and water ad libitum. The animals were divided into four groups: (1) SAL (0.9% saline, *n* = 6), (2) SAL + calcitriol (0.9% saline + calcitriol, *n* = 6), (3) CP (cisplatin 5mg/kg, *n* = 8) and (4) CP + calcitriol (cisplatin 5mg/kg + calcitriol, *n* = 8). Calcitriol (6 ng/day, Calcijex, Abbvie Laboratories, North Chicago, IL, USA) or vehicle (0.9% NaCl) was administered using miniosmotic pumps (model 2004, Alzet, Cupertino, CA, USA) implanted subcutaneously under isoflurane anesthesia (Cristalia, Brazil). Calcitriol or vehicle supplementation was started two weeks before the injection of CP and was maintained five days later, corresponding to the period evaluated. The dose of calcitriol treatment was selected according to previous studies [27,28,60]. None of the rats died after cisplatin administration. All animals were used in the study.

### 4.2. Renal Function Studies

On the fourth day after CP injection, the animals were placed in metabolic cages for 24 h to collect urine samples. On the fifth day after CP injection, the animals were anesthetized (xylazine 0.1 mL/100 g and ketamine 0.05 ml/100 g, i.p.), the aorta was cannulated and blood samples were collected. Renal function was assessed using 24 h urine and blood samples. Plasma and urinary creatinine were determined by the colorimetric method using picric acid as a chromogen [61]. Urinary and plasma sodium were analyzed using the ion-selective electrode quantification technique (9180 Electrolyte Analyzer, Roche Diagnostics GmbH, Mannheim, Germany, 2004). Fractional sodium excretion was calculated by dividing sodium clearance by creatinine clearance. The results of the plasmatic and urinary creatinine quantification were used to determine the glomerular filtration rate (GFR).

### 4.3. Serum 25 Hydroxyvitamin D (25 OHD) Levels

We assessed 25(OHD) with a direct competitive test based on the chemiluminescence principle (CLIA) (DiaSorin, Liaison®, Saluggia, Italy); this test was performed in the clinical analysis laboratories at the School of Medicine of Ribeirao Preto Hospital and Clinics, which participates in national and international quality assurance certification.

### 4.4. Histological Studies

Histological sections (4 μm thick) were stained using Masson’s Trichrome and examined under light microscopy (Axion Vision Rel. 4.3; Zeiss, Oberkochen, Germany). Tubulointerstitial changes interstitial infiltration of inflammatory cells, atrophy of the cells of the renal tubules and dilation of the tubular lumen were evaluated.

Lesions in the renal outer medulla were graded [53] on a scale of 0–4 as follows (0 = normal; 0.5 = small focal areas; 1 = involvement of <10% of the renal outer medulla; 2 = 10–25%; 3 = 25–75%; 4 = extensive damage involving more than 75% of the renal outer medulla). Thirty grid fields measuring 0.1 mm^2^ were evaluated in the renal outer medulla of each kidney (Axion version 4.8.3, Zeiss, Oberkochen, Germany), and the mean values per kidney were calculated.

### 4.5. Immunohistochemical Studies

For immunohistochemical analysis, kidney sections were deparaffinized and hydrated. Nonspecific antigen binding was blocked by incubation for 20 min with normal goat serum. The sections were incubated with anti-ED1 (1:1000, Serotec, Oxford, UK), anti-vimentin (1:500, Dako Corporation, Glostrup, Denmark), anti-PCNA (1:1000, Sigma Chemical Company, St. Louis, USA), α-SMA (1:100, Dako Corporation, Glostrup, Denmark), cubilin (1:200, Santa Cruz Biotechnology, Santa Cruz, CA, USA) and anti-aminopeptidase P (JG12) (1:1000, Bioscience, San Diego, USA) antibodies for one hour at room temperature for reaction of the primary antibody. The avidin–biotin–peroxidase complex (Vector Laboratories Inc., Burlingame, USA) was used to detect the reaction product. The color reaction, in turn, was developed with DAB (3,3′-diaminobenzidine (Sigma Chemical Company, Burlington, VT, USA) and nickel chloride in the presence of H_2_O_2_. Counterstaining of the sections was then performed with methyl green, which was followed by dehydration and mounting. The immunoperoxidase staining for ED1 and PCNA was determined by counting the number of positive cells in the renal outer medulla. JG12 was determined by the number of positive peritubular capillaries in the outer renal medulla. The reaction to cubilin was evaluated by counting the number of intact tubules with a cubilin-marked brush border in the tubules of the renal outer medulla. Vimentin and α-SMA were semiquantitatively graded in the outer medulla, and the mean score per kidney was calculated. The scores depended on the percentage of a grid field showing positive staining as follows: 0 = absent or <5% staining, 1 = 5–25%, 2 = 25–50%, 3 = 50–75% and 4 = >75% staining. Thirty consecutive fields (0.1 mm^2^ each) for the outer medulla were evaluated. The average score per kidney was calculated. All fields were analyzed under 400× magnification.

### 4.6. Western Blot Studies

To perform the Western blot, the nonperfused kidney tissues were homogenized in lysis buffer: Tris-HCL (50 Mm, pH 7.4), NaCl (150 mM), triton X-100 (1%), dodecyl sulfate sodium (SDS; 0.1%), aprotinin (1 μg/mL), leupeptin (1 μg/mL), sodium fluoride (25 Mm), sodium ethylenediamine acid tetrapyrophosphate (1mM), sodium fluoride (25 Mm) and ethylene diamine tetraacetic acid (0.001 M EDTA, pH 8) at 4 °C [56]. Bradford’s method (1976) measured the proteins in lysate samples. Renal expressions of eNOS, p-eNOS, EC-SOD, VEGF, VEGFR2, CXCR4, CD34, vimentin and VDR were evaluated [62]. Renal lysate samples containing 30, 60 or 90 μg of protein were solubilized in sample buffer, heated to 100 °C for 5 min and then applied to a 10% or 12% polyacrylamide gel. After the run, the samples were transferred from the gel to a nitrocellulose membrane. The membranes were incubated or not for one hour in a 5% molecular blocking buffer. Subsequently, the membranes were washed in TBSt and incubated with the primary antibodies: anti-NOS3 (1/200, polyclonal, Santa Cruz Biotechnology, Santa Cruz, CA, USA), anti-p-eNOS (1/200, Santa Cruz Biotechnology), anti-EC-SOD (1/500, polyclonal, Santa Cruz Biotechnology), anti-VEGF (1/500, polyclonal, Santa Cruz Biotechnology), anti-VEGFR2 (1/200, polyclonal, Santa Cruz Biotechnology), anti-CXCR4 (1/1000, polyclonal, Santa Cruz Biotechnology), anti-VDR (1/500, polyclonal, Santa Cruz Biotechnology), anti-Vimentina (1/500, monoclonal, Dako, Denmark), anti-CD34 (1/400, polyclonal, BIOss, Boston, MA, USA) and/or anti-GAPDH (1/1000, monoclonal, Sigma Chemical Company, Burlington, NJ, USA). The membranes were washed and incubated with anti-rabbit (1/10,000) or mouse (1/10,000) anti-IgG secondary antibodies linked to peroxide (Dako, Denmark) for one hour at room temperature. The result of the reaction was detected with luminol and captured in a computerized system. The intensity of the identified bands was quantified by densitometry with NIH Image J software (Research Services Branch, USA) and reported in arbitrary units. Protein estimates were evaluated by the Bradford method.

### 4.7. ELISA Studies

Levels of IL-1β and IL-10 were measured in kidney tissue samples, which were stored at −70 °C until analysis. The content was determined using ELISA kits according to the manufacturer’s guidelines (Alpco, Keewaydin Drive, Tulane, USA; Pierce, Waltham, MA, USA, respectively). IL-1β and IL-10 values are reported in in picograms/milligrams (pg/mg) of protein.

### 4.8. Statistical Analyses

For data with a normal distribution, analysis of variance and the Newman–Keuls multiple comparison tests were applied. For data not normally distributed, the Kruskal–Wallis nonparametric test followed by Dunn’s posttest was used. The Kolmogorov–Smirnov trial investigated the normality of the dependent variables. Data are presented as mean ± SEM. GraphPad Prism version 9.0 for Windows (GraphPad Software 9.0, San Diego, CA, USA) was used to perform the statistical analysis and subsequent graph construction. Statistical significance was established at *p* < 0.05.

## Figures and Tables

**Figure 1 ijms-23-15877-f001:**
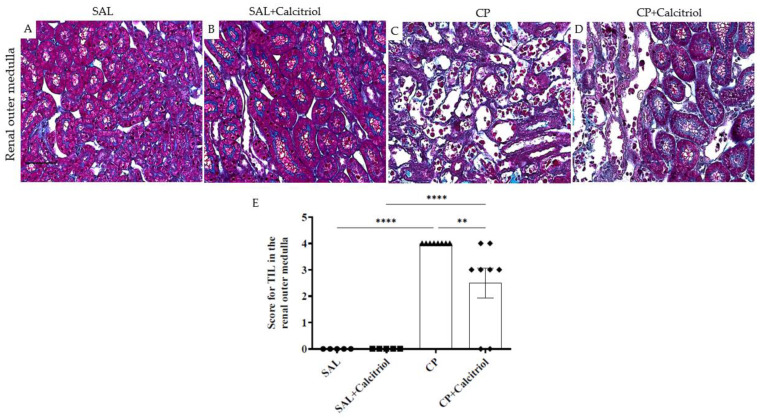
Histological sections stained with Masson’s Trichrome representative of the renal outer medulla of groups SAL (**A**), SAL + calcitriol (**B**), CP (**C**) and CP + calcitriol (**D**) (bar represents 50 µm). The score for TIL in the renal outer medulla (**E**) of all experimental groups. Data are expressed as mean ± SEM (*n* = 5–8 for each group). ** *p* < 0.01; **** *p* < 0.0001 ×400.

**Figure 2 ijms-23-15877-f002:**
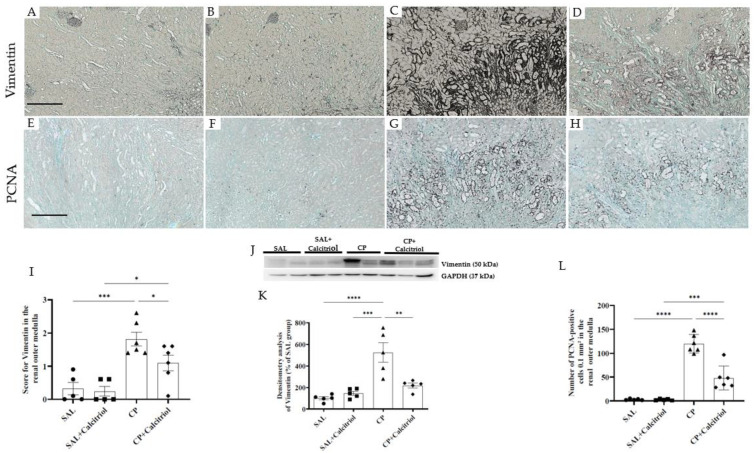
Immunolocalization of vimentin (**A**–**D**) and PCNA (**E**–**H**) in the renal outer medulla of the SAL (**A**,**E**), SAL + calcitriol (**B**,**F**), CP (**C**,**G**) and CP + calcitriol (**D**,**H**) groups. The bar indicates 200 μm. The score for vimentin (**I**) and the number of PCNA+ cells (**L**) in the renal outer medulla in the different groups. Western blot analysis of vimentin and GAPDH (**J**) in the renal tissue from all experimental groups (SAL, SAL + calcitriol, CP, CP + calcitriol). Vimentin densitometry (**K**). The densitometric ratio between vimentin and GAPDH was calculated, and data are expressed in comparison with the control group, with the mean control value (±SEM) designated as 100% and expressed as mean ± SEM (*n* = 5–8 for each group). Blots are representative images of independent experiments. * *p* < 0.05; ** *p* < 0.01; *** *p* < 0.001; **** *p* < 0.0001: magnification, ×100.

**Figure 3 ijms-23-15877-f003:**
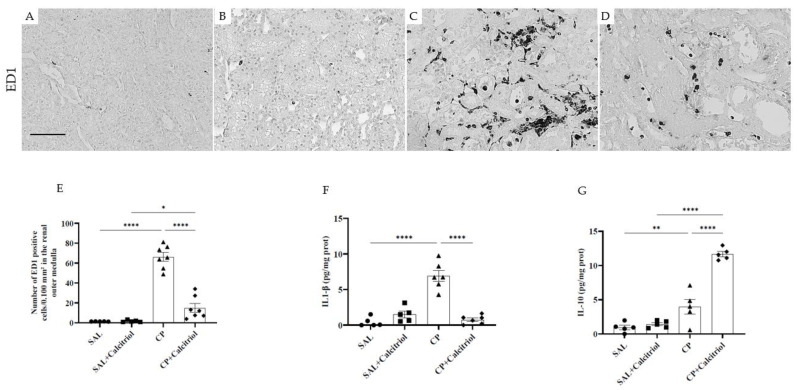
Immunolocalization of ED1-positive cells (macrophages) in the outer renal medulla of the SAL (**A**), SAL + calcitriol (**B**), CP (**C**) and CP + calcitriol (**D**) groups (bar represents 50 μm). The number of ED1-positive cells in all experimental groups’ outer renal medulla (**E**). Values are given as the mean ± SEM. Renal tissue levels of IL (interleukin)-1β (**F**) and IL (interleukin)-10 (**G**) from control (SAL and SAL + calcitriol) and experimental (CP and CP + calcitriol) groups. Data are expressed as mean ± SEM (*n* = 5–8 for each group). * *p* < 0.05; ** *p* < 0.01; **** *p* < 0.0001: magnification, ×400.

**Figure 4 ijms-23-15877-f004:**
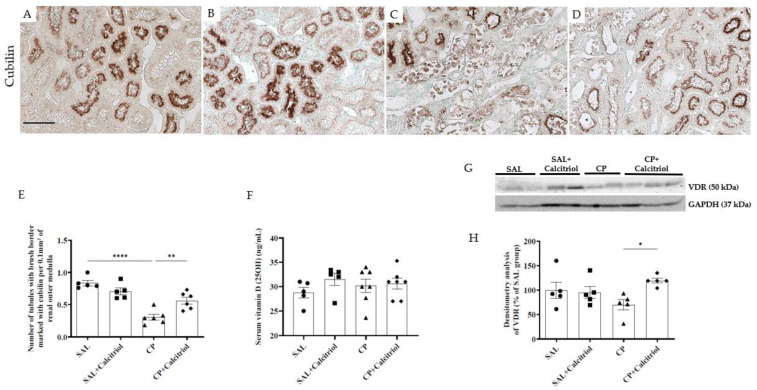
Immunolocalization of cubilin in the renal outer medulla of the SAL (**A**), SAL + calcitriol (**B**), CP (**C**) and CP + calcitriol (**D**) groups (bar indicates 50 μm). The number of tubules with a brush border marked with cubilin (**E**) from all experimental groups (SAL, SAL + calcitriol, CP, CP + calcitriol). Serum levels of 25(OH) of all experimental groups (SAL, SAL + calcitriol, CP, CP + calcitriol) (**F**). Western blot analysis of vitamin D receptor (VDR) and GAPDH (**G**) in the renal tissue from all experimental groups (SAL, SAL + calcitriol, CP, CP + calcitriol). VDR densitometry (**H**). The densitometric ratio between VDR and GAPDH was calculated, and data are expressed in comparison with the control group, with the mean control value (±SEM) designated as 100% and expressed as mean ± SEM (*n* = 5–8 for each group). Blots are representative images of independent experiments. * *p* < 0.05; ** *p* < 0.01; **** *p* < 0.0001: magnification, ×400.

**Figure 5 ijms-23-15877-f005:**
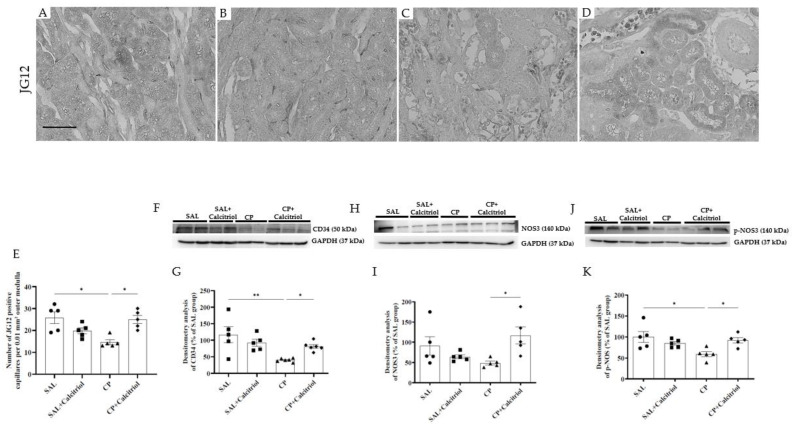
Immunolocalization of JG12 (**A**–**D**) in the renal outer medulla of the SAL (**A**), SAL + calcitriol (**B**), CP (**C**) and CP + calcitriol (**D**) animals. The bar indicates 50 μm. The number of JG12-positive capillaries in all experimental groups’ outer renal medulla (**E**). Western blot analysis of CD34 (**F**), NOS3 (**H**) and p-NOS (**J**) in the renal tissue from all experimental groups (SAL, SAL + calcitriol, CP, CP + calcitriol). Densitometry of CD34 (**G**), NOS3 (**I**) and p-NOS (**K**). The densitometric ratio between CD34, NOS3, p-NOS and GAPDH was calculated, and the data are expressed in comparison with the control group, with the mean control value (±SEM) designated as 100% and expressed as mean ± SEM (*n* = 5–8 for each group). Blots are representative images of independent experiments. * *p* < 0.05; ** *p* < 0.01.

**Figure 6 ijms-23-15877-f006:**
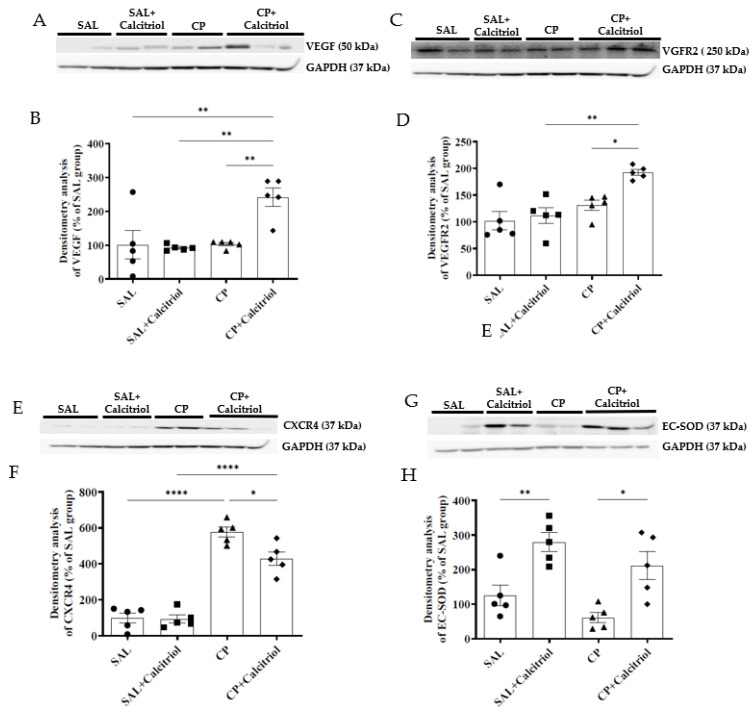
Western blot analysis of VEGF (**A**), VEGFR2 (**C**), CXCR4 (**E**) and EC-SOD (**G**) in the renal tissue from all experimental groups (SAL, SAL + calcitriol, CP, CP + calcitriol). Densitometry of VEGF (**B**), VEGFR2 (**D**), CXCR4 (**F**) and EC-SOD (**H**). The densitometric ratio between VEGF, VEGFR2, CXCR4, EC-SOD and GAPDH was calculated, and the data are expressed in comparison with the control group, with the mean control value (±SEM) designated as 100% and expressed as mean ± SEM (*n* = 5–8 for each group). Blots are representative images of independent experiments. * *p* < 0.05; ** *p* < 0.01; **** *p* < 0.0001.

**Table 1 ijms-23-15877-t001:** Plasma creatinine (Pcreat), fractional sodium excretion (FE_Na+_) and glomerular filtration rate (GFR) 5 days after injection of CP or vehicle of the SAL (*n* = 6), SAL + calcitriol (*n* = 6), CP (*n* = 8) and CP + calcitriol (*n* = 8) groups.

Group	SAL	SAL + Calcitriol	CP	CP + Calcitriol
Pcreat (mg %)	0.54 ± 0.03	0.45 ± 0.01	6.18 ± 0.74 ***; ●●●	1.89 ± 0.43 ■■■
FE+ Na (%)	0.25 ± 0.5	0.26 ± 0.02	2.23 ± 1.14 ***; ●●●	1.48 ± 0.24 **; ●
GFR (mL/min^1^ 100 g^1^)	0.79 ± 0.03	1.46 ± 0.21	0.03 ± 0.01 ***; ●●●	0.52 ± 0.15 **; ●●; ■

Data are expressed as mean ± S.E.M. ** *p* < 0.01, *** *p* < 0.001 vs. SAL; ● *p* < 0.05, ●● *p* < 0.01, ●●● *p* < 0.001 vs. SAL + calcitriol.; ■ *p* < 0.05, ■■■ *p* < 0.001 vs. CP.

## Data Availability

Not applicable.

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
