# Peer review of "Calcitriol Reduces the Inflammation, Endothelial Damage and Oxidative Stress in AKI Caused by Cisplatin"

_ijms, 2022, doi:10.3390/ijms232415877_

Round 1
Reviewer 1 Report
Oliveira et.al demonstrated that calcitriol reduces the inflammation, endothelial damage and
oxidative Stress in AKI caused by cisplatin. Other groups have reported that vitamin D or calcitriol play protective role in cisplatin induced AKI via the inflammatory and endothelial injury process. This study focused more on proximal tubular injury, oxidative stress, inflammation and vascular injury. Based on the novelty and the quality of the study, it is not suitable for publishing on International Jounrnal of Modecular Sciences (the current IF is 6.208).
Please see the comments below:
1. Figure1. is there a specific result that the author showed the structure of the renal outer medulla? Cisplatin is more retained in the proximal tubule and induces proximal tubular injury. PAS staining to show the cortical region of the kidney will be a better readout for cisplatin-induced AKI.
2. For the western blot in Fig2, Figure4, Figure5, and Figure6, the CP group only has 2 samples, it is hard to quantify the difference among those groups.
Author Response
Review 1
We are grateful for the extensive revision of the manuscript and the suggestions made, and we are replying below to each point raised.
- Figure 1 is there a specific result that the author showed the structure of the renal outer medulla? Cisplatin is more retained in the proximal tubule and induces proximal tubular injury. PAS staining to show the cortical region of the kidney will be a better readout for cisplatin-induced AKI.
Cisplatin nephrotoxicity is widely studied in rats and it has been shown that cisplatin accumulation in the kidney involving selective damage to S3 of the proximal tubule located in the outer stripe of the outer medulla. (Nokasa et al., 1992).
Nosaka K., Nakada J., Endou H. Cisplatin-induced alterations in renal structure, ammoniagenesis and gluconeogenesis of rats. Kidney International, 1992, 41,73—79.
- For the western blot in Fig2, Figure4, Figure5, and Figure6, the CP group only has 2 samples, it is hard to quantify the difference among those groups.
The membranes shown in figures 2, 4, 5 and 6 are representative of the graph, to quantify the difference between the groups, the same number of samples was used, with n=5 samples per group. This information is described in the figure legend.
Reviewer 2 Report
Dear authors you will find attached my comments.

Author Response
We are grateful for the extensive revision of the manuscript and the suggestions made, and we are replying below to each point raised.
Q1: Line 33. Please add the definition of AKI (eg KDIGO definition)
Thank you for the suggestion. We added the information.
AKI is a clinical condition characterized by an abrupt fall in renal function reversible, decline in the glomerular filtration rate (GFR) and the accumulation of metabolic waste [5]
Q2 : Table 1 Please change the table configuration (without symbols)
We use symbols to show statistical differences.
Q3 : Figure 1 : I think this figure is not clear. It should be improved
The quality of the image was improved.
Q4 Figure 2 : You should improve the figure. Αlso, images should be in color.
The quality of the image was improved.
Q5 Line 278 please rephrase
Thank you for the suggestion. We rephrased.
Reviewer 3 Report
In this paper, Oliveria et al. investigated the role of calcitriol in a rat model of AKI.
I believe that this paper is easily readable.
Major
-
Methods. The authors claim that “Calcitriol or vehicle supplementation was started two weeks before injection of CP and was maintained five days later, corresponding to the period evaluated”. Why did the author decide to start the calcitriol/vehicle supplementation two weeks before? How was this timing determined?
-
Table 1. Please provide the exact number of rats used, here and in the Methods section.
-
Table 1. The SAL+calcitriol group has a GFR almost double of SAL alone. Can the author provide an explanation for this?
-
Table 1. Renal function in CP and CP+calcitriol was assessed 5 days after CP injection. Why did the authors choose this time point? This should be carefully explained. How can the authors be sure that the improvement observed is elicited by calcitriol administration and not by renal recovery itself? Is it possible to provide other time points for the assessment of kidney function?
-
How many rats died after CP administration?
-
The Discussion lacks a Limitations section.
Minor:
-
Page 1 line 36-38: this sentence is not clear; consider rephrasing.
-
Page 2 line 51: what is RO?
Author Response
Review 3
We are grateful for the extensive revision of the manuscript and the suggestions made, and we are replying below to each point raised.
- The authors claim that “Calcitriol or vehicle supplementation was started two weeks before injection of CP and was maintained five days later, corresponding to the period evaluated”. Why did the author decide to start the calcitriol/vehicle supplementation two weeks before? How was this timing determined?
The time was defined based on previous studies where vitamin D supplementation was initiated before the induction of renal injury. In addition to adding time for the animals to recover from the surgery, before inducing the injury.
Shena Q., Bib X., Lingc L., Dingb W. 1,25-Dihydroxyvitamin D3 Attenuates Angiotensin II-Induced Renal Injury by Inhibiting Mitochondrial Dysfunction and Autophagy. Cell Physiol Biochem 2018, 51, 1751-1762.
Mohammed M. A., Aboulhoda, B. E., Mahmoud, R. H. Vitamin D attenuates gentamicin-induced acute renal damage via prevention of oxidative stress and DNA damage. Human and Experimental Toxicology, 2019, 38, 321–335.
- Table 1. Please provide the exact number of rats used, here and in the Methods section.
Thanks for the suggestion, the exact number of animals was added in Table 1 and in the methods.
- Table 1. The SAL+calcitriol group has a GFR almost double of SAL alone. Can the author provide an explanation for this?
Experimental studies with animal models have demonstrated renoprotective effects of active vitamin D, effects mediated by its receptor (VDR), regulating multiple pathways, including renal structural proteins (Li, 2012). Second, we measured plasma creatinine levels with routine clinical laboratory methods, but not with high-performance liquid chromatography, a more reliable method for evaluating plasma creatinine levels.
Li, Y.C. Vitamin D: Roles in renal and cardiovascular protection. Curr. Opin. Nephrol. Hypertens. 2012, 21, 72–79.
- Table 1. Renal function in CP and CP+calcitriol was assessed 5 days after CP injection. Why did the authors choose this time point? This should be carefully explained. How can the authors be sure that the improvement observed is elicited by calcitriol administration and not by renal recovery itself? Is it possible to provide other time points for the assessment of kidney function?
The chosen evaluation time was based on previous studies in our laboratory. We chose to evaluate 5 days after CP administration, as this corresponds to the peak of renal injury and the mechanisms involved are activated and well established. We believe that the observed improvement is provoked by the administration of Calcitriol, since the recovery is only observed in the CP group that received calcitriol. If it were for renal recovery itself, we would observe the same effect in the CP group.
- How many rats died after CP administration?
None of the rats died after cisplatin administration. All animals were used in the study. This information has been included in the methods section.
- The Discussion lacks a Limitations section.
Thanks for the suggestion. A limitations section has been included in the discussion
Minor:
- Page 1 line 36-38: this sentence is not clear; consider rephrasing.
Thanks for the suggestion. The sentence has been reworded.
Before:
“AKI is a clinical condition characterized by an abrupt fall in renal function reversible, decline in the glomerular filtration rate (GFR) and the accumulation of metabolic waste [5].”
After:
“AKI is characterized by an abrupt drop in renal function, decline in glomerular filtration rate (GFR) and accumulation of metabolic waste [5].”
- Page 2 line 51: what is RO
The RO noted on page 2 line 51 was a typo, actually it was supposed to be ROS, short for reactive oxygen species. It has already been corrected in the text.
Round 2
Reviewer 1 Report
1. The novelty of the paper is largely compromised since the protective role of calcitriol in cisplatin-induced AKI via the inflammatory and endothelial injury process has been reported.
2. The author did not perform PAS staining in the kidney which is a typical method to evaluate the injury of kidney structure in the kidney.
3. At least three samples to show the change in protein level from each group are needed.
Without improving the novelty and using the appropriate methodology to describe the kidney injury and a scientific way to quantify the change of protein expression, this study is hard to be accepted by the International Jounrnal of Modecular Sciences (the current IF is 6.208).
Author Response
Review 1
We are grateful for the extensive revision of the manuscript and the suggestions made, and we are replying below to each point raised.
- The novelty of the paper is largely compromised since the protective role of calcitriol in cisplatin-induced AKI via the inflammatory and endothelial injury process has been reported.
This study explored the effects of calcitriol administration on proximal tubular injury, oxidative stress, inflammation and vascular injury observed in CP-induced AKI.
- The author did not perform PAS staining in the kidney which is a typical method to evaluate the injury of kidney structure in the kidney.
Most articles that work with renal parenchyma microscopy use 3 stains; they are Hematoxylin-Eosin (HE), Masson's Trichrome and Periodic Acid Schiff (PAS). PAS staining is used to highlight glomerular lesions; whereas tubular lesions are best demonstrated with Masson's trichrome stain. In addition, scar formation and sclerotic lesions are also better appreciated using Masson's trichrome. CP-induced AKI is characterized by an intense acute tubular lesion, so we chose Masson’s trichrome as the stain to highlight this lesion.
- At least three samples to show the change in protein level from each group are needed.
As previously stated, 5 samples from each group were used for analysis of protein expression by Western Blot, the membrane presented in the results section is only representative of each group, the entire membranes were sent in the supplementary material. The graphs were plotted with the individual value of the animals in each group, making it possible to observe the behavior of each one individually.
Reviewer 2 Report
Dear authors
I have observed your corrections and find them satisfactory.However, I think the diagrams need to be improved more.
Author Response
Review 2
We are grateful for the extensive revision of the manuscript and the suggestions made, and we are replying below to each point raised.
I have observed your corrections and find them satisfactory. However, I think the diagrams need to be improved more.
We are grateful for the suggestions made.
Reviewer 3 Report
Much improved. I have no other issues to raise.
Round 3
Reviewer 1 Report
The author claimed that this study is more focused on the proximal tubule, however, this is not a proximal tubule-specific injury model. The novelty of this study is largely compromised since the protective role of vitamin D or calcitriol in cisplatin-induced AKI via the inflammatory and endothelial injury process has been reported.
In terms of the aim and scope of the International Journal of Mechanical Sciences (Only original, innovative, and novel papers will be considered for publication in the International Journal of Mechanical Sciences), this study is not suitable for publishing.